civil engineering/engineering geology

deep hole mining, blasting-roof, plastic bearing, virtual work principle, numerical simulation

**Author for correspondence:**
Yaguang Qin
e-mail: csuqyg@163.com

# Plastic limit bearing calculation of blasting-roof in deep hole mining and its applications

Wei Wang[1], Zhouquan Luo[1], Yaguang Qin[1] and Jun Xiang[2]

[1]School of Resources and Safety Engineering, Central South University, Changsha 410083, People's Republic of China
[2]Fankou Lead-Zinc Mine, Shaoguan 512325, People's Republic of China

A plastic bearing calculation method for a blasting-roof is proposed to solve the problem of determining the blasting-roof thickness in deep hole mining. A mechanical analysis model for the plastic bearing was built for the typical boundary conditions of blasting-roofs. The external and internal work of the blasting-roof are equal under the plastic limit state through calculation. The limit bearing formulae of blasting-roofs under various boundary conditions were derived based on the principle of virtual work. A Vertical Crater Retreat stope was taken as the object, and the safe blasting-roof thickness was determined to be 6 m using the derived formula (considering the safety coefficient). A numerical model of stope was constructed using the Surpac-Flac3D technique, while the blasting-roof stability was simulated under different thicknesses. Variations in the simulated indexes (stress and plastic zone volume) prove that the theoretical calculations are reliable. The plastic bearing calculation method can provide a new method to determine the blasting-roof thickness in deep hole mining.

## 1. Introduction

With the gradual depletion of mineral resources in shallow areas, multiple mines have now entered the stage of deep mining. The unique environment in deep regions (high stress, high temperature, high water pressure, and strong disturbance) requires mining innovation, with safety and efficiency having been the main targets of mining enterprises [1]. In other countries, deep hole mining is widely used due to its high efficiency and safety [2]. This approach, specifically the Vertical Crater Retreat (VCR) mining method, was introduced in China

in the 1980s. After decades of development, the VCR method has become the epitome of efficient and safe mining. The present status of the VCR method is summarized in four points: (i) ordinary emulsion explosives replace ammonium nitrate explosives to reduce blasting costs. The blasting charge structure evolved from a single-layer globular to a multiple-layer column [3], which significantly improved the blasting efficiency; (ii) down-upside funnel blasting was widely used in early VCR mining approaches, which had a limited free surface and low blasting efficiency. The kerf full-hole lateral blasting and kerf sectional lateral blasting techniques were subsequently invented to improve the blasting efficiency and have since been frequently used in VCR mining [4,5]; (iii) with the development of mining equipment, the cave height in deep hole mines can often reach 30 m, with some even exceeding 100 m. For instance, the cave height using the VCR method in the Anqing copper mine has reached 120 m; (iv) the application of a visual remote-control scraper transforms the stope bottom structure from a funnel and trench to being flat, which significantly reduces the mining workload and bottom pillar losses.

Due to its representation and progressiveness, deep hole mining has been a well-studied topic among mining scholars. Monjezi & Dehghani [6] established a neural network between the blasting parameters and vibration effect to reasonably select the blasting parameters. Liu et al. [7] simulated the blasting process with different charge weights and coupling coefficients and chose an optimized charge structure through simulations. Shim et al. [8] optimized the blasting design based on the spatial distribution of rock factors, which significantly reduced the blasting fragmentation and costs. Li et al. [9] applied the 'small resistance blasting', 'hole bottom reversed blasting', and 'non electric blasting network' techniques in a uranium mine and showed improvements to the blasting efficiency. Liu et al. [10] conducted directional focused-energy blasting tests and monitored the blasting process using a high-speed camera and analysed the crack development and stress evolution to verify the feasibility of the approach. Industrial experiments, theoretical analyses, monitoring, and numerical simulations promote the scientific rational and feasibility of deep hole mining. However, deep hole mining is extremely complex because of the blast instantaneity [4]. There are still some parameters that are difficult to determine in practice, including the blasting-roof thickness (final area in the kerf blasting), which directly influences the efficiency and safety of the kerf blasting, making approaches for its rational determination important.

Currently, there is no systematic research available concerning blasting-roof thickness quantification; therefore, we propose a method to determine this quantity. First, a mechanical analysis model for the plastic limit bearing is built. Second, the external and internal work of the blasting-roof under the plastic limit state are calculated. The limit bearing formulae of a blasting-roof under various boundary conditions are derived based on the principle of virtual work. Taking a VCR stope as the object, the safe blasting-roof thickness is determined from the derived formula that considers the safety coefficient, and the calculation results are verified through numerical simulations. This study aims to provide a new way to determine the blasting-roof thickness in deep hole mines.

## 2. Deep hole mining method

The VCR mining method is shown in figure 1.

The top of the stope contains the drilling chamber while the bottom is the ore pass chamber. The blasting holes are made using a down-the-hole drill. Kerf blasting is arranged at one end of the stope to generate a free surface and compensation space. The last step of the kerf blasting is the blasting-roof, where the remaining ore is caved from the lateral blasting. The caved ore is transported through the visual remote-control scraper. The blasting-roof is the final work platform of the kerf blasting, so its stability is extremely important for workers and equipment. Nevertheless, if the blasting-roof is too thick, the kerf blasting efficiency will decrease and the generated free surface incline will be uneven. Thus, the blasting-roof thickness should be rationally determined.

## 3. Plastic bearing analysis of blasting-roof

The plastic limit analysis is often used to analyse the bearing capacity of elastic-plastic structures [11]. The advantage of this method is that only the ultimate bearing state of the structure is considered during the analysis, and the calculation results are the same as for the elastic-plastic analysis [12]. Here, the plastic limit analysis is adopted to analyse the limit bearing capacity of the blasting-roof.

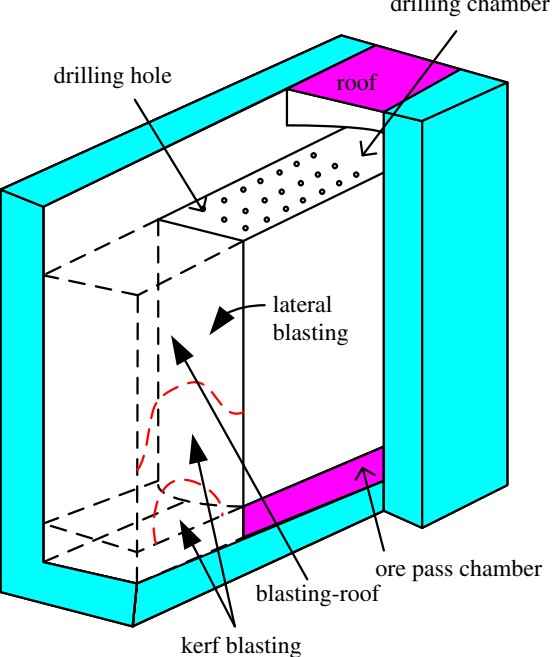

**Figure 1.** Deep hole mining method.

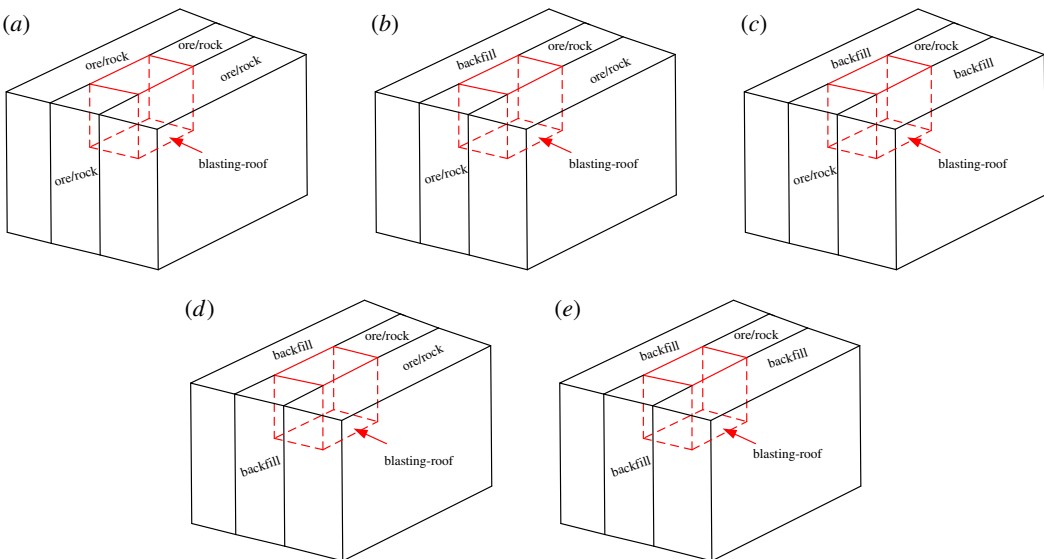

**Figure 2.** Boundary conditions of blasting-roof: (*a*) four fixed; (*b*) three fixed one simply supported; (*c*) opposite fixed opposite simply supported; (*d*) adjacent fixed adjacent simply supported; (*e*) three simply supported one fixed.

## 3.1. Boundary conditions

The kerf full-hole lateral blasting and the kerf sectional lateral blasting are often used in deep hole mining operations. According to the actual mining layout, there are five boundary conditions of blasting-roofs, which are shown in figure 2. Specifically, these include four sides of ore/rock, one side backfill and three sides ore/rock, two sides backfill and two sides ore/rock (two different forms), and three sides backfill and one side ore/rock. Some assumptions are made to simplify the theoretical analysis as listed below.

(i) The ore/rock boundary (owing to its high strength) is fixed, while the backfill boundary (low strength) is a simply-supported boundary.
(ii) The elastic deformation in the blasting-roof is relatively small.

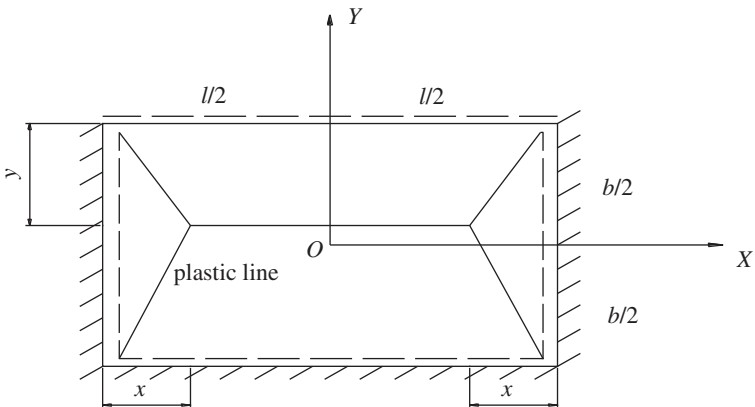

**Figure 3.** Bearing analysis model of blasting-roof.

(iii) During the damage process, plastic lines are only generated along fixed boundaries.
(iv) Under the plastic limit bearing state, the blasting-roof is divided into several blocks by the plastic lines.

Based on these assumptions, the boundary conditions for the blasting-roof can be concluded as four fixed sides, three fixed sides and one simply-supported side, opposite simply-supported sides and opposite fixed sides, adjacent fixed sides and adjacent simply-supported sides, three simply-supported sides and one fixed side.

## 3.2. Mechanical analysis

The calculation of the plastic limit bearing is illustrated using the example boundary condition of three fixed sides and one simply-supported side. The mechanical analysis model of the plastic limit bearing was built (figure 3) using the previous assumptions. The length of the blasting-roof was set as $l$ in m, and the width was set as $b$ in m. The distance between the intersection point of the plastic lines and the eastern–western boundary was set as $x$ in m, while the distance between the intersection point and the north–south boundary was $y$ in m.

The maximum virtual displacement of the blasting-roof was set as $\delta$ in m, the limit bending moment was set as $M_s$ in N•m, and the external force was set as $q$ in N. By definition [13], the external work ($w_e$) is given by:

$$w_e = \sum_{j-1}^{n} \left( \iint_{A_j} qw(x,y)\mathrm{d}A_j \right) = q \sum_{j-1}^{n} V_j, \tag{3.1}$$

where $n$ is the number of blocks divided by the plastic lines and $V_j$ is the volume of block $j$ in m³. Expanding equation (3.1), the detailed expression of the external work is:

$$w_e = q\left[\tfrac{1}{6}\delta(b-y)(3l-4x) + \tfrac{1}{6}\delta y(3l-4x) + \tfrac{1}{6}\delta bx + \tfrac{1}{6}\delta x\right]. \tag{3.2}$$

The internal work ($w_i$) is the product of three parts.

$$w_i = \sum_{j-1}^{n} (M_{sj}L_j\theta_j), \tag{3.3}$$

where $\theta_j$ is the normal angle of block $j$ in degrees, $M_{sj}$ is the projection of $M_s$ on the plastic rotation axis in N•m, and $L_j$ is the total projection length in m. Expanding equation (3.3), the detailed expression for the internal work is:

$$w_i = 2M_s l\frac{\delta}{b-y} + M_s l\frac{\delta}{y} + 4M_s b\frac{\delta}{x}. \tag{3.4}$$

When the blasting-roof was in a limited damage state, the external and internal forces of the blasting-roof remained in balance [14]. According to the principle of virtual work, the external work is equal to

**Table 1.** Plastic limit bearing expressions of the blasting-roof.

| boundary condition | bearing expression ($q_s$) | limit bearing ($q_m$) | parameter note |
|---|---|---|---|
| three fixed one simply-supported | $\dfrac{6M_s[lx(b+y)+4by(b-y)]}{(3l-2x)(b-y)bxy}$ | $\dfrac{6M_s k_1}{(3-2\sqrt{2})(3-2K_2 k^2)K_2 l^2 k^2}$ | $k=b/l$  $K_1=8\sqrt{\left(272-192\sqrt{2}\right)+\left(18-12\sqrt{2}\right)/k^2}$  $K_2=8\sqrt{2}-12+K_1$ |
| four fixed | $\dfrac{24M_s(b^2+2lx)}{(3l-2x)b^2 x}$ | $\dfrac{48M_s(1+K_3)}{(3-k^2 K_3)l^2 k^2 K_3}$ | $k=b/l$  $K_3=\sqrt{1+(3/k^2)}-1$ |
| adjacent fixed adjacent simply-supported | $\dfrac{6M_s[lx_1 x_2(b+y)+by(2x_2+x_1)(b-y)]}{[3l-2(x_1+x_2)](b-y)byx_1 x_2}$ | $\dfrac{6M_s\left[K_4+6\left(3-2\sqrt{2}\right)\right]}{(3-2\sqrt{2})(3-2K_4 k^2)K_4 l^2 k^2}$ | $k=b/l$  $K_4=12\sqrt{2}-18+3\sqrt{\left(6-4\sqrt{2}/k^2\right)+\left(68-48\sqrt{2}\right)}$ |
| opposite fixed opposite simply-supported | $\dfrac{12M_s(8lx+b^2)}{(3l-2x)l^2 x}$ | $\dfrac{384M_s(1+K_5)}{(12-K_5 k^2)l^2 k^2}$ | $k=b/l$  $K_5=\sqrt{1+(12/k^2)}-1$ |
| three simply-supported one fixed | $\dfrac{6M_s[lx(b+y)+2by(b-y)]}{(3l-2x)(b-y)bxy}$ | $\dfrac{6M_s K_7}{(3-2\sqrt{2})(3-2K_6 k^2)K_6 l^2 k^2}$ | $k=b/l$  $K_6=8\sqrt{\left(68-68\sqrt{2}\right)+\left(9-6\sqrt{2}\right)/k^2}$  $K_7=4\sqrt{2}-6+K_6$ |

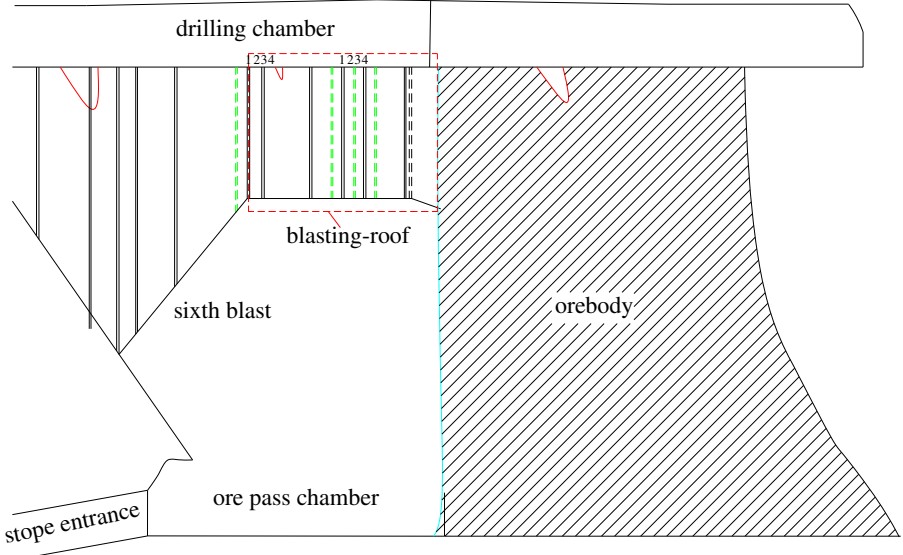

**Figure 4.** Blasting-roof design of stope.

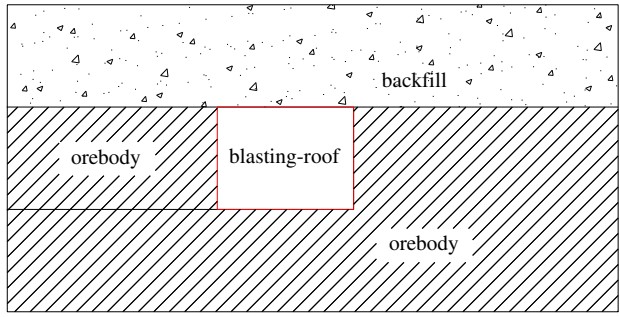

**Figure 5.** Boundary condition of blasting-roof.

the internal work.

$$w_e = w_i. \tag{3.5}$$

According to equations (3.2)–(3.5), the plastic bearing expression of the blasting-roof ($q_s$) is derived as:

$$q_s = \frac{6M_s[lx(b+y) + 4by(b-y)]}{(3l-2x)(b-y)bxy}. \tag{3.6}$$

The quantities $x$ and $y$ were solved under the assumptions of $d_q/d_x = 0$ and $d_q/d_y = 0$.

$$\left.\begin{array}{l} x = lK_1k^2 \\ y = (\sqrt{2}-1)b \end{array}\right\} \tag{3.7}$$

Combining equations (3.6) and (3.7), the plastic limit bearing ($q_m$) of the blasting-roof was obtained as:

$$\left.\begin{array}{l} q_m = \dfrac{6M_sK_1}{(3-2\sqrt{2})(3-2K_2k^2)K_2l^2k^2} \\[2mm] M_s = \dfrac{\sigma_t h^2}{6} \\[2mm] K_1 = 8\sqrt{\left(272 - 192\sqrt{2}\right) + \left(18 - 12\sqrt{2}\right)/k^2} \\[2mm] K_2 = 8\sqrt{2} - 12 + K_1 \end{array}\right\} \tag{3.8}$$

where $k = b/l$, $\sigma_t$ is the tensile strength of the ore in MPa, and $h$ is the blasting-roof thickness in m.

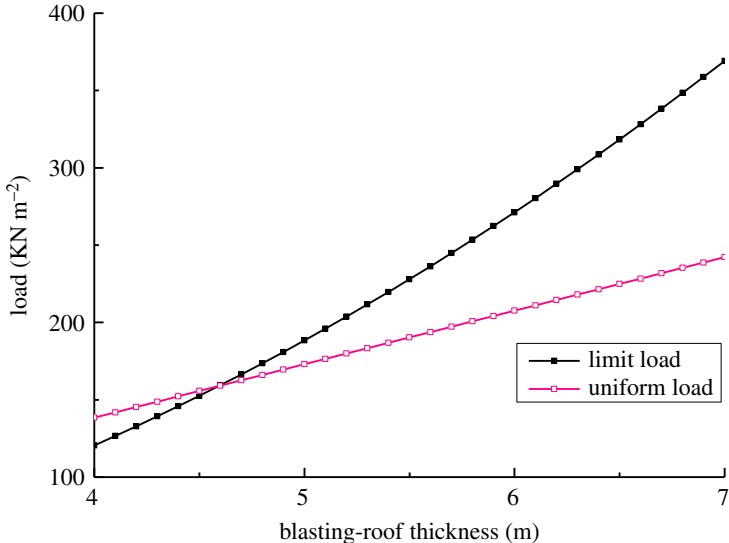

**Figure 6.** Variation of blasting-roof bearing capacity.

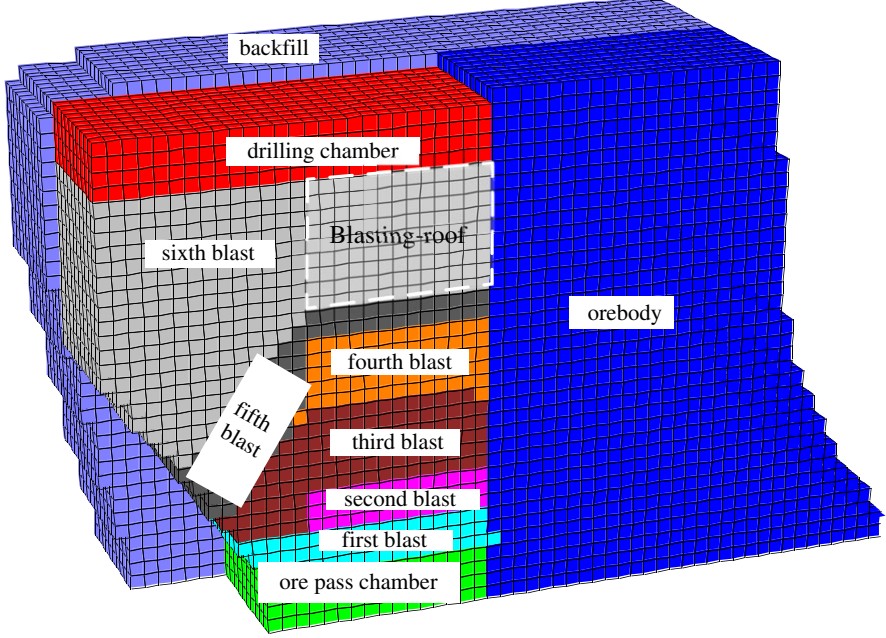

**Figure 7.** Numerical model of stope.

## 3.3. Plastic limit bearing of the blasting-roof

Similarly, the plastic limit bearings of the blasting-roof under the remaining boundary conditions were also derived, as shown in table 1.

# 4. Engineering application

## 4.1. Stope condition

The object VCR mining stope is located at a depth of 600 m in a lead-zinc mine. The north side of the stope is backfill, the southern and eastern side is ore and the west side is rock. The average dip and thickness of the orebody are 40° and 23 m, respectively, and the Protodyakonov coefficient of the ore and rock are 4–17 and 8–10, respectively.

(*a*)

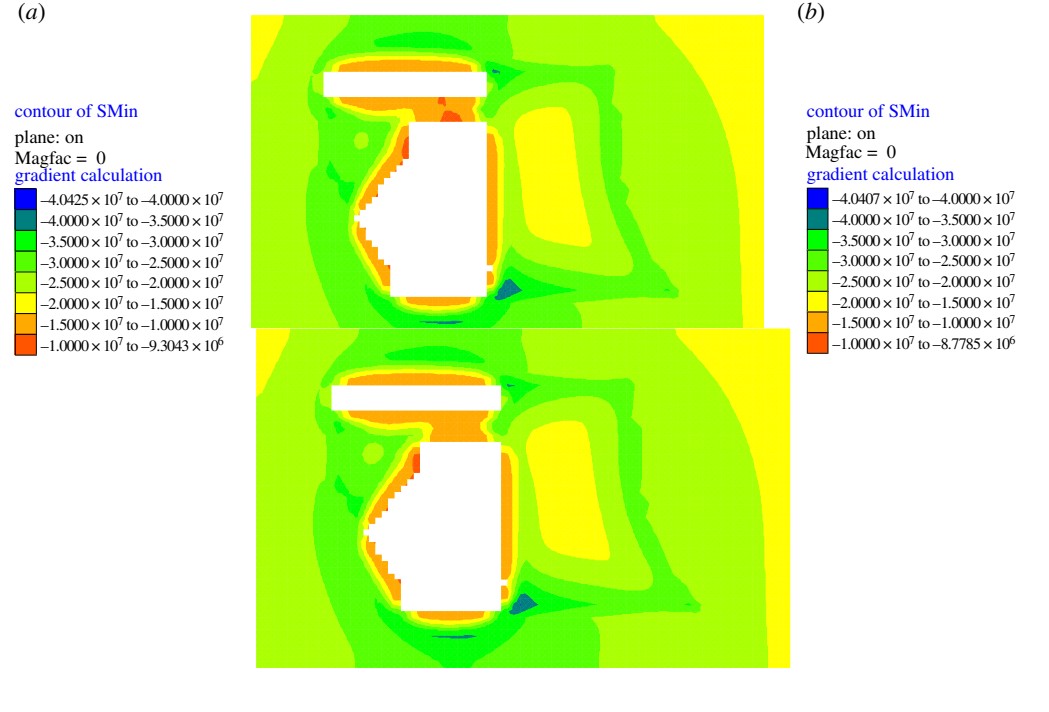

(*b*)

(*c*)

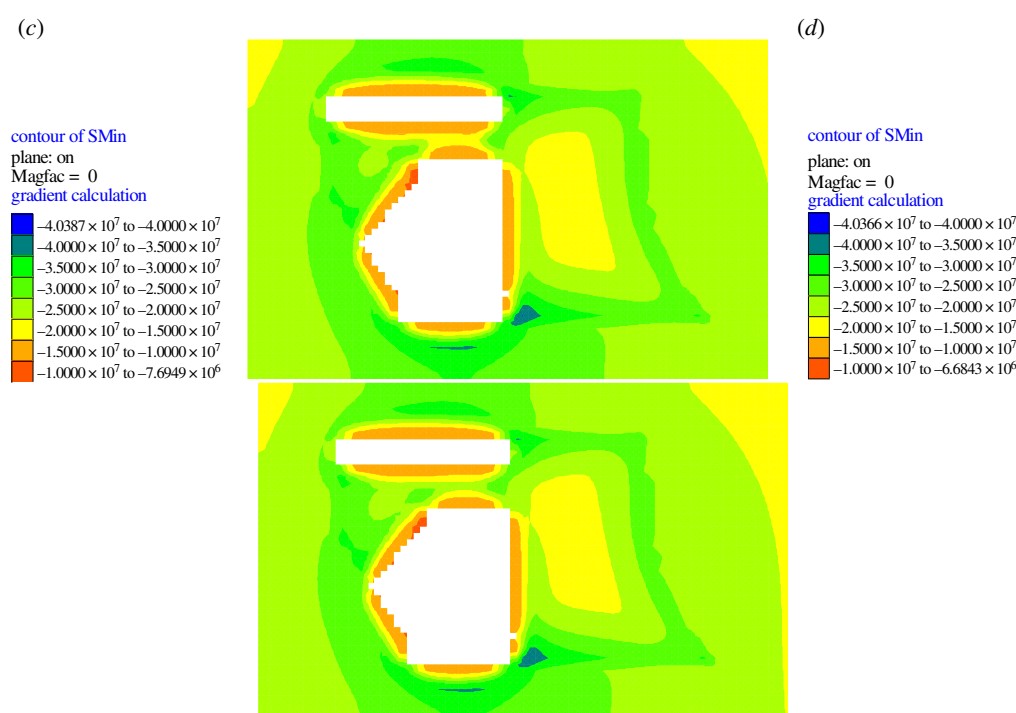

(*d*)

**Figure 8.** Stress distribution with different blasting-roof thickness: (*a*) 4 m; (*b*) 5 m; (*c*) 6 m; (*d*) 7 m.

The stope is planned to have six blastings, and the design and boundary conditions for the blasting-roof are shown in figures 4 and 5.

## 4.2. Determination of safe blasting-roof thickness

The boundary condition can be classified as three fixed sides and one simply-supported side. According to the derived formula for the plastic limit bearing, the calculation parameters are: $l = 13$ m, $b = 10$ m, $k = 0.77$, $\sigma_t = 2.32$ MPa and $\gamma = 34.6$ KN m$^{-3}$. The plastic bearing of the blasting-roof is calculated using equation (3.8), and the upper uniform load is calculated with $\gamma h$. The bearing capacity variation

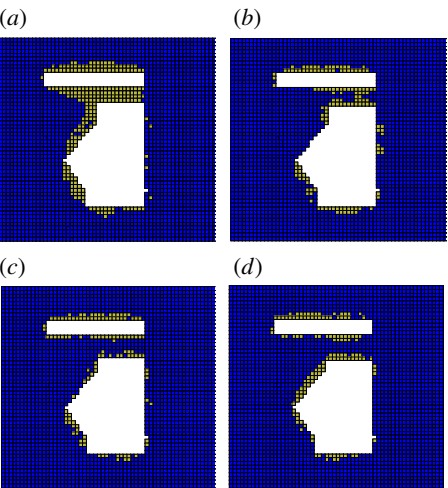

**Figure 9.** Plastic zone distribution with different blasting-roof thickness: (*a*) 4 m; (*b*) 5 m; (*c*) 6 m; (*d*) 7 m.

of the blasting-roof is shown in figure 6. It is seen that the blasting-roof bearing capacity grows with its thickness and reached a limit failure state at a thickness of 4.6 m. When the thickness exceeded this value, the limit bearing capacity was higher than the upper uniform load, putting the blasting-roof in a favourable state. Correspondingly, the blasting-roof was completely damaged when the thickness was less than 4.6 m. Therefore, the minimum blasting-roof thickness from the mechanical analysis is determined to be 4.6 m.

In practice, a series of factors (underground water, abrupt stress, engineering disturbance, etc.) have unpredictable impacts on the blasting-roof stability, and only the limit failure state is considered in the theoretical analysis. So, the calculation result should be revised by considering the safety coefficient (a similar underground mine had a safety coefficient value of 1.2–1.5). When the rock around the stope is relatively stable, the groundwater and engineering disturbances have a minimal influence on the stope, and the safety factor is set to 1.2. Considering that one side of the object stope is backfill, the influence of groundwater and engineering disturbances on the stope is very small, so the safety coefficient value is set to 1.3. Therefore, the safe blasting-roof thickness is finally determined to be 6 m.

# 5. Numerical simulation analysis

## 5.1. Numerical model

The Surpac-Flac$^{3D}$ modelling technique was used to construct the numerical model of the object stope. First, a 3D solid model of the stope was built using Surpac. Second, the solid model was converted to a block grid model and exported as the centroid file. This file was imported using Access and edited with SQL operations. Thus, the original data was transformed into four point coordinates ($P_0$–$P_3$) and exported as a text file (available for Flac$^{3D}$). Considering the boundary effect, the model range was set as ($x$, $y$, $z$) in m from (2670 to 2770, 8500 to 8600, −500 to −650). Figure 7 shows the internal work of constructed numerical models.

## 5.2. Simulation results analysis

From the previous calculations, the minimum blasting-roof thickness was determined to be 4.6 m using the mechanical analysis, and the blasting-roof thickness considering the safety coefficient was 6 m. To examine the reliability of the theoretical analysis, four groups of blasting-roof thicknesses (4, 5, 6 and 7 m) were selected for the numerical simulations. The mechanical response characteristics under different blasting-roof thicknesses are shown in figures 8 and 9.

As shown in figure 8, the mining process influences the stress distribution of the blasting-roof, which increases with smaller thicknesses. The stress value in the blasting-roof decreased under the four thickness conditions, reflecting the process that the blasting-roof damage weakens the rock-mass strength. When the thickness was 6 and 7 m, the stress-weakened zone in the blasting-roof was

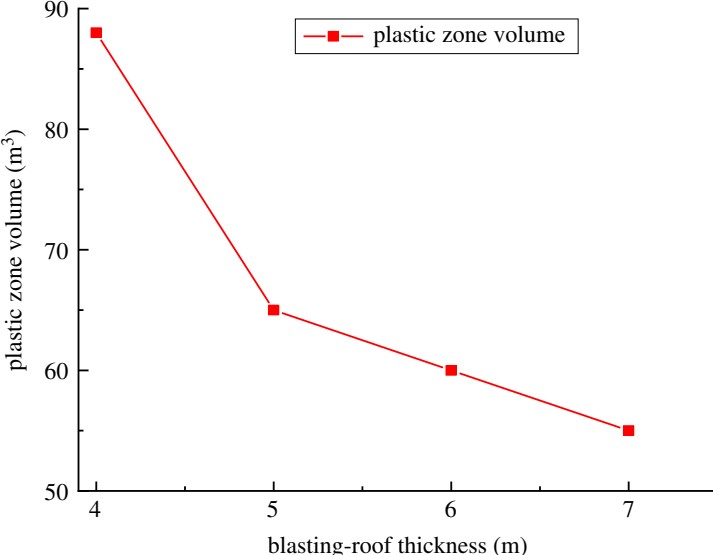

**Figure 10.** Variation of plastic zone volume.

relatively small. The stress-weakened zone evolved and covered most of the area of the blasting-roof when the thickness reduced to 4 and 5 m.

The variations in the plastic zone (figure 9) indicate that the blasting-roof damage evolved from a small to a large scale as the thickness reduced. Similarly, when the thickness was 6 or 7 m, there was only a small plastic zone in the blasting-roof. When the thickness reduced to 4 or 5 m, the plastic zone expanded to most areas. Key monitoring points were established to record the plastic zone volume in the numerical simulation, and the results are shown in figure 10.

The plastic zone volume decreased significantly when the blasting-roof thickness reduced from 5 to 4 m. This further proves that the blasting-roof nearly reached the limit damage state when the thickness was less than 5 m. In summary, the numerical simulation results confirm the reliability of the theoretical analyses. Specifically, the blasting-roof is completely damaged when the thickness decreases to below 4.6 m and is safe when reaching 6 m.

# 6. Conclusion

(i) A new method to determine the blasting-roof thickness in deep hole mines was proposed. The mechanical analysis model of the plastic limit bearing was built based on the five typical boundary conditions. The external and internal work of the blasting-roof are equal under the plastic limit state in the calculations, and the plastic limit bearing formulae for the blasting-roof were derived.

(ii) Taking the VCR stope in a lead-zinc mine as an example, the minimum blasting-roof thickness was determined to be 4.6 m as derived from the developed formulae, and the thickness when considering the safety coefficient was 6 m. Numerical simulations of the blasting-roof stability proved the reliability of the theoretical analysis. The plastic limit bearing calculations can provide a new method to determine the blasting-roof thickness in deep hole mines.

Ethics. Permission to collect samples of the stope in the Fankou lead-zinc mine was granted to Wei Wang and Jun Xiang, and the study was reviewed and approved by the School of Resources and Safety Engineering, Central South University. Zhouquan Luo and Yaguang Qin collected the measured goaf model of the stope.

Data accessibility. Data are deposited in the Dryad Digital Repository: https://doi.org/10.5061/dryad.hk0tf00 [15].

Authors' contributions. W.W. and Z.L. carried out the theoretical analysis; Y.Q. participated in the data analysis and collected the field data; J.X. collected the field data; W.W. and Y.Q. drafted the manuscript. All authors gave final approval for publication.

Competing interests. The authors declare no competing interests.

Funding. The work reported in this paper was supported by The National Key R&D Program of China during the Thirteenth Five-year Plan Period (2017YFC0602901); National Natural Science Foundation of China (51274250); and The Fundamental Research Funds for the Central Universities of Central South University (2017zzts204).

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
