## [Reviewer comments · Royal Society Open Science]

Review History

RSOS-190074.R0 (Original submission)

Review form: Reviewer 1

Is the manuscript scientifically sound in its present form?

Yes

Are the interpretations and conclusions justified by the results?

Yes

Is the language acceptable?

Yes

Is it clear how to access all supporting data?

Yes

Do you have any ethical concerns with this paper?

No

Have you any concerns about statistical analyses in this paper?

No

Recommendation?

Accept with minor revision (please list in comments)

Comments to the Author(s)

In the process of deep hole blasting, the thickness of blasting roof will directly affect the blasting effect and operation safety. By establishing the analytical model of plastic bearing capacity of blasting roof, the authors put forward the plastic ultimate bearing capacity expression of blasting roof, which is suitable for different boundary conditions. It provides a new idea for determining the thickness of blasting roof in deep hole mining.

However, there are still some problems about their work. The problems are listed below:

(1) In Section 3.1, the boundary conditions are simplified. After simplification, the calculation results will be affected. How to eliminate these effects?

(2) As shown in figures 6, the author points out that "When the thickness exceeds 4.6m, limit bearing capacity is higher than upper uniform load, so the blasting-roof is in a favorable state". When the thickness is 4.6m, the overlying load may have local stress concentration, which exceeds the ultimate load. How to deal it?

(3) As shown in figures 8 and 9, when the thickness of blasting roof is 6 meters or 7 meters, the plastic failure and the stress weakening range are small. Does the thickness of blasting roof satisfy the safety requirement as long as it is more than 6 meters?

(4) There are some grammar errors in the article.

In summary, minor repairs are recommended.

Review form: Reviewer 2

Is the manuscript scientifically sound in its present form?

Yes

Are the interpretations and conclusions justified by the results?

Yes

Is the language acceptable?

Yes

Is it clear how to access all supporting data?

Yes

Do you have any ethical concerns with this paper?

No

Have you any concerns about statistical analyses in this paper?

No

Recommendation?

Accept with minor revision (please list in comments)

Comments to the Author(s)

The Authors proposed the plastic limit bearing calculation method of blasting-roof. The results are nicely presented and the conclusions can be supported by the study results. It has important guiding significance for deep hole mining. The reviewer recommends a minor revision prior to the publication of the paper in the journal Royal Society Open Science. Below shows the minor comments on the manuscript:

(1) Some assumptions are made for the sake of simplifying theoretical analysis. How to deal with the affect of assumptions on the calculation results? Whether the calculation result is verified in engineering application?

The variables in formula 1-8 should be given units to facilitate readers to verify the calculation.

(3) Please check the spell and grammatical errors.

(4) Please figure out the position of Fig. 2 in the description. In Fig 10, it is recommended that the thickness of blasting roof varies from small to large rather than from large to small.

(5) The authors should explain the specific method of getting the safety coefficient value. And, the safety coefficient value should be confirmed in the paper.

(6) Please figure out the difference between the external work of blasting-roof and the internal work blasting-roof. In the paper (Page 4, Line 47), the external work is equal to internal work. Therefore, the expression in the Abstract and Conclusion (i.e., The external work and internal work of blasting-roof under plastic limit state were calculated) should be more specific.

Decision letter (RSOS-190074.R0)

24-Apr-2019

Dear Dr Yaguang,

The editors assigned to your paper ("Plastic Limit Bearing Calculation of Blasting-Roof in Deep Hole Mining and Its Application") have now received comments from reviewers. We would like you to revise your paper in accordance with the referee and Associate Editor suggestions which can be found below (not including confidential reports to the Editor). Please note this decision does not guarantee eventual acceptance.

Please submit a copy of your revised paper before 17-May-2019. Please note that the revision deadline will expire at 00.00am on this date. If we do not hear from you within this time then it will be assumed that the paper has been withdrawn. In exceptional circumstances, extensions may be possible if agreed with the Editorial Office in advance. We do not allow multiple rounds of revision so we urge you to make every effort to fully address all of the comments at this stage. If deemed necessary by the Editors, your manuscript will be sent back to one or more of the original reviewers for assessment. If the original reviewers are not available, we may invite new reviewers.

- Data accessibility

If you wish to submit your supporting data or code to Dryad (<http://datadryad.org/>), or modify your current submission to dryad, please use the following link:
<http://datadryad.org/submit?journalID=RSOS&manu=RSOS-190074>

- Competing interests

- Authors' contributions

- Acknowledgements

- Funding statement

Kind regards,

Andrew Dunn

on behalf of Prof R. Kerry Rowe (Subject Editor)

Associate Editor's comments:

The referees have identified a number of concerns with your paper that require revisions before the manuscript may be considered further for publication. Please ensure you fully incorporate the changes requested, and provide a point-by-point response to the reviewers. In addition, it has been observed that the paper would benefit from language polishing (<https://royalsociety.org/journals/authors/language-polishing/>) before resubmission. Please ensure you seek appropriate advice and provide evidence of having done so when you resubmit.

Comments to Author:

Reviewers' Comments to Author:

Reviewer: 1

Comments to the Author(s)

In the process of deep hole blasting, the thickness of blasting roof will directly affect the blasting effect and operation safety. By establishing the analytical model of plastic bearing capacity of blasting roof, the authors put forward the plastic ultimate bearing capacity expression of blasting roof, which is suitable for different boundary conditions. It provides a new idea for determining the thickness of blasting roof in deep hole mining.

However, there are still some problems about their work. The problems are listed below:

(1) In Section 3.1, the boundary conditions are simplified. After simplification, the calculation results will be affected. How to eliminate these effects?

(2) As shown in figures 6, the author points out that "When the thickness exceeds 4.6m, limit bearing capacity is higher than upper uniform load, so the blasting-roof is in a favorable state". When the thickness is 4.6m, the overlying load may have local stress concentration, which exceeds the ultimate load. How to deal it?

(3) As shown in figures 8 and 9, when the thickness of blasting roof is 6 meters or 7 meters, the plastic failure and the stress weakening range are small. Does the thickness of blasting roof satisfy the safety requirement as long as it is more than 6 meters?

(4) There are some grammar errors in the article.

In summary, minor repairs are recommended.

Reviewer: 2

Comments to the Author(s)

The Authors proposed the plastic limit bearing calculation method of blasting-roof. The results are nicely presented and the conclusions can be supported by the study results. It has important guiding significance for deep hole mining. The reviewer recommends a minor revision prior to the publication of the paper in the journal Royal Society Open Science. Below shows the minor comments on the manuscript:

(1) Some assumptions are made for the sake of simplifying theoretical analysis. How to deal with the affect of assumptions on the calculation results? Whether the calculation result is verified in engineering application?

The variables in formula 1-8 should be given units to facilitate readers to verify the calculation.

(3) Please check the spell and grammatical errors.

(4) Please figure out the position of Fig. 2 in the description. In Fig 10, it is recommended that the thickness of blasting roof varies from small to large rather than from large to small.

(5) The authors should explain the specific method of getting the safety coefficient value. And, the safety coefficient value should be confirmed in the paper.

(6) Please figure out the difference between the external work of blasting-roof and the internal work blasting-roof. In the paper (Page 4, Line 47), the external work is equal to internal work. Therefore, the expression in the Abstract and Conclusion (i.e., The external work and internal work of blasting-roof under plastic limit state were calculated) should be more specific.

Author's Response to Decision Letter for (RSOS-190074.R0)

See Appendix A.

RSOS-190074.R1 (Revision)

Review form: Reviewer 1

Is the manuscript scientifically sound in its present form?

Yes

Are the interpretations and conclusions justified by the results?

Yes

Is the language acceptable?

Yes

Is it clear how to access all supporting data?

Yes

Do you have any ethical concerns with this paper?

No

Have you any concerns about statistical analyses in this paper?

No

Recommendation?

Accept as is

Comments to the Author(s)

The responses to the reviewers' comments are correct and reasonable.
I suggest to accept this manuscript.

Review form: Reviewer 2

Is the manuscript scientifically sound in its present form?

Yes

Are the interpretations and conclusions justified by the results?

Yes

Is the language acceptable?

Yes

Is it clear how to access all supporting data?

Yes

Do you have any ethical concerns with this paper?

No

Have you any concerns about statistical analyses in this paper?

No

Recommendation?

Accept as is

Comments to the Author(s)

I am satisfied with the reply of the author. I have no other comments.

Decision letter (RSOS-190074.R1)

10-May-2019

Dear Dr Yaguang:

On behalf of the Editors, I am pleased to inform you that your Manuscript RSOS-190074.R1 entitled "Plastic Limit Bearing Calculation of Blasting-Roof in Deep Hole Mining and Its Application" has been accepted for publication in Royal Society Open Science subject to minor revision in accordance with the referee suggestions. Please find the referees' comments at the end of this email.

The reviewers and Subject Editor have recommended publication, but also suggest some minor revisions to your manuscript. Therefore, I invite you to respond to the comments and revise your manuscript.

- Ethics statement

- Data accessibility

If you wish to submit your supporting data or code to Dryad (<http://datadryad.org/>), or modify your current submission to dryad, please use the following link:
<http://datadryad.org/submit?journalID=RSOS&manu=RSOS-190074.R1>

- Competing interests

- Authors' contributions

- Acknowledgements

- Funding statement

Because the schedule for publication is very tight, it is a condition of publication that you submit the revised version of your manuscript before 19-May-2019. Please note that the revision deadline will expire at 00.00am on this date. If you do not think you will be able to meet this date please let me know immediately.

Supplementary files will be published alongside the paper on the journal website and posted on

the online figshare repository (<https://figshare.com>). The heading and legend provided for each supplementary file during the submission process will be used to create the figshare page, so please ensure these are accurate and informative so that your files can be found in searches. Files on figshare will be made available approximately one week before the accompanying article so that the supplementary material can be attributed a unique DOI.

on behalf of Prof R. Kerry Rowe (Subject Editor)
openscience@royalsociety.org

Associate Editor Comments to Author:

You were asked by the Editors to seek advice from a language editing service (<https://royalsociety.org/journals/authors/language-polishing/>) to help improve the clarity of your manuscript. You have not provided any evidence of having received such advice. Please ensure that you have your paper edited by a qualified language editor prior to submitting the revision - if you do not provide any evidence that you have done so, we'll return the paper to you.

Reviewer comments to Author:
Reviewer: 2

Comments to the Author(s)
I am satisfied with the reply of the author. I have no other comments.

Reviewer: 1

Comments to the Author(s)
The responses to the reviewers' comments are correct and reasonable.
I suggest to accept this manuscript.

Author's Response to Decision Letter for (RSOS-190074.R1)

See Appendices B & C.

Decision letter (RSOS-190074.R2)

23-May-2019

Dear Dr Yaguang,

I am pleased to inform you that your manuscript entitled "Plastic Limit Bearing Calculation of Blasting-Roof in Deep Hole Mining and Its Applications" is now accepted for publication in Royal Society Open Science.

on behalf of Prof R. Kerry Rowe (Subject Editor)
openscience@royalsociety.org

Appendix A

Reply to the editor's and reviewers' comments

Dear editor and reviewers:

We sincerely thank you for all your comments on our manuscript (ID RSOS-190074). Our studies assessed the Plastic Limit Bearing Calculation of Blasting-Roof in Deep Hole Mining and Its Application. In the current revision, we have modified the manuscript following your comments. The revised text is highlighted in red in the revised manuscript. Point-by-point responses to the reviewers' comments are listed below.

I look forward to hearing from you soon. Since the manuscript had been polished by Elsevier before its submission, so they were invited to revise it again.

Yours sincerely,
Yaguang Qin

Associate Editor's comments:

The referees have identified a number of concerns with your paper that require revisions before the manuscript may be considered further for publication. Please ensure you fully incorporate the changes requested, and provide a point-by-point response to the reviewers. In addition, it has been observed that the paper would benefit from language polishing (<https://royalsociety.org/journals/authors/language-polishing/>) before resubmission. Please ensure you seek appropriate advice and provide evidence of having done so when you resubmit.

Thank you for your kind suggestion.

Reviewer 1:

In the process of deep hole blasting, the thickness of blasting roof will directly affect the blasting effect and operation safety. By establishing the analytical model of plastic bearing capacity of blasting roof, the authors put forward the plastic ultimate bearing capacity expression of blasting roof, which is suitable for different boundary conditions. It provides a new idea for determining the thickness of blasting roof in deep hole mining. However, there are still some problems about their work. The problems are listed below:

(1) In Section 3.1, the boundary conditions are simplified. After simplification, the calculation results will be affected. How to eliminate these effects?

Reply: Considering that the boundary conditions are simplified, in Section 4.3 the calculation result is revised with safety coefficient (safety coefficient value: 1.2~1.5).

These effects can be effectively eliminated by multiplying the safety coefficient.

(2) As shown in figures 6, the author points out that “When the thickness exceeds 4.6m, limit bearing capacity is higher than upper uniform load, so the blasting-roof is in a favorable state”. When the thickness is 4.6m, the overlying load may have local stress concentration, which exceeds the ultimate load. How to deal it?

Reply: We thank the reviewer’s comment. The theoretical calculation shows that the thickness of the broken roof is 4.6m. The overlying load may have local stress concentration that exceeds the ultimate load. Therefore, the theoretical value is revised with safety coefficient to ensure the reliability of thickness of blasting roof.

(3) As shown in figures 8 and 9, when the thickness of blasting roof is 6 meters or 7 meters, the plastic failure and the stress weakening range are small. Does the thickness of blasting roof satisfy the safety requirement as long as it is more than 6 meters?

Reply: When the thickness of blasting roof is more than 6 meters it can satisfy the safety requirement. However, if the blasting-roof is too thick, kerf blasting efficiency will decrease and generated free surface incline to be uneven. So the reasonable thickness of blasting roof is 6m

(4) There are some grammar errors in the article.

Reply: We are sorry for the mistake. We have carefully checked the grammar errors in the article and revised them.

Reviewer 2:

The Authors proposed the plastic limit bearing calculation method of blasting-roof. The results are nicely presented and the conclusions can be supported by the study results. It has important guiding significance for deep hole mining. The reviewer recommends a minor revision prior to the publication of the paper in the journal Royal Society Open Science. Below shows the minor comments on the manuscript:

(1) Some assumptions are made for the sake of simplifying theoretical analysis. How to deal with the affect of assumptions on the calculation results? Whether the calculation result is verified in engineering application?

Reply: Considering that the assumptions are made for the sake of simplifying theoretical analysis, in Section 4.3 the calculation result is revised with safety coefficient (safety coefficient value: 1.2~1.5). These effects can be effectively eliminated by multiplying the safety coefficient. The calculation result had been verified in engineering application

(2)The variables in formula 1-8 should be given units to facilitate readers to verify the

calculation.

Reply: We are thankful for the reviewer's kind suggestion. We have given the units of variables in formula 1-8.

(3) Please check the spell and grammatical errors.

Reply: We are sorry for the mistake. We have carefully checked the spell and grammar errors in the article and revised them.

(4) Please figure out the position of Fig. 2 in the description. In Fig 10, it is recommended that the thickness of blasting roof varies from small to large rather than from large to small.

Reply: Thank you for the suggestion. We have figured out the position of Fig.2 in the manuscript and revised the Fig.10.

(5) The authors should explain the specific method of getting the safety coefficient value. And, the safety coefficient value should be confirmed in the paper.

Reply: Thank you for reminding us. When the surrounding rock around the slope is very stable, groundwater and engineering disturbance have little influence on the slope, and the safety factor is set to 1.2. Considering that one side of the object slope is backfill, the influence of groundwater and Engineering Disturbance on the slope is very small, so the safety coefficient value is set to 1.3.

(6) Please figure out the difference between the external work of blasting-roof and the internal work blasting-roof. In the paper (Page 4, Line 47), the external work is equal to internal work. Therefore, the expression in the Abstract and Conclusion (i.e., The external work and internal work of blasting-roof under plastic limit state were calculated) should be more specific.

Reply: We are thankful for the reviewer's kind suggestion. The external work mainly includes blasting vibration and mechanical work. Internal work is mainly the work done by gravity. The expression in the Abstract and Conclusion about external work and internal work are revised.

Appendix B

Reply to the editor' s and reviewers' comments

Dear editor and reviewers:

We sincerely thank you for all your comments on our manuscript (ID RSOS-190074). Our studies assessed the Plastic Limit Bearing Calculation of Blasting-Roof in Deep Hole Mining and Its Applications. In the current revision, we have modified the manuscript following your comments.

I look forward to hearing from you soon. Since the manuscript had been polished by Elsevier before its submission, so they were invited to revise it again.

Yours sincerely,

Ya-guang QIN

Associate Editor's comments:

You were asked by the Editors to seek advice from a language editing service (<https://royalsociety.org/journals/authors/language-polishing/>) to help improve the clarity of your manuscript. You have not provided any evidence of having received such advice. Please ensure that you have your paper edited by a qualified language editor prior to submitting the revision - if you do not provide any evidence that you have done so, we'll return the paper to you.

Reply: Thank you for your kind suggestion. We have sought advice from the language editing service (<https://royalsociety.org/journals/authors/language-polishing/>) to help improve the clarity of our manuscript.

Reviewer 1:

The responses to the reviewers' comments are correct and reasonable.
I suggest to accept this manuscript.

Reply: We are thankful for the reviewer's kind suggestion.

Reviewer 2:

I am satisfied with the reply of the author. I have no other comments.

Reply: We are thankful for the reviewer's kind suggestion.

Appendix C

EDITORIAL CERTIFICATE

This document certifies that the manuscript below was edited for correct English language usage, grammar, punctuation and spelling by qualified native English speaking editors at The Charlesworth Group.

Paper Title:

Plastic limit bearing calculation of blasting-roof in deep hole mining and its application

Author:

yaguang qin

Date certificate issued:

May 15, 2019

cwauthors.com